# Machine learning assisted composition design of high-entropy Pb-free relaxors with giant energy-storage

Xingcheng Wang[1,4], Ji Zhang [2,4], Xingshuai Ma[1,4], Huajie Luo[1], Laijun Liu [3], Hui Liu [1]✉ & Jun Chen [1]✉

The high-entropy strategy has emerged as a prevalent approach to boost capacitive energy-storage performance of relaxors for advanced electrical and electronic systems. However, exploring high-performance high-entropy systems poses challenges due to the extensive compositional space. Herein, with the assistance of machine learning screening, we demonstrated a high energy-storage density of 20.7 J cm$^{-3}$ with a high efficiency of 86% in a high-entropy Pb-free relaxor ceramic. A random forest regression model with key descriptors based on limited reported experimental data were developed to predict and screen the elements and chemical compositions of high-entropy systems. Following basic experiments, a $(Bi_{0.5}Na_{0.5})TiO_3$-based high-entropy relaxor characterized by fine grains, weakly-coupled and small-sized polar clusters was identified. This resulted in a near-linear polarization behavior and an ultrahigh breakdown strength of 95 kV mm$^{-1}$. Further, this high-entropy realxor presented a high discharge energy density of 7.7 J cm$^{-3}$ under discharge rate of about 27 ns, along with superior temperature and fatigue stability. Our results present the data-driven model for efficiently exploring high-performance high-entropy relaxors, demonstrating the potential of machine learning in developing relaxors.

Electrostatic ceramic energy-storage capacitors play a crucial role in advanced pulsed power systems and are integral to various applications such as electric vehicles, portable electronic devices, and power grids[1–7]. With ongoing technological progress, there is a growing necessity to develop the next generation of dielectric capacitors with high energy-storage density ($W_{rec}$) and efficiency ($\eta$) to fulfill the requirements of miniaturization and integration[8–13]. Therefore, developing dielectric materials with high energy-storage performance has become a prominent and critical area of research in recent years[14–18].

Typically, the $W_{rec}$ of a dielectric is primarily influenced by its electric-field ($E$) induced polarization curve and breakdown strength ($E_B$), while the $\eta$ is correlated with the hysteresis ($H$) from the irreversible polarization response. Among various dielectrics, perovskite-structured relaxor ferroelectrics (RFEs) are stand out as promising candidates due to their high polarization ($P_m$) and low $H$[19,20]. Chemical compositional design, involving such as doping, substitution, and mixing with different components is a key strategy for modifying the heterogeneous polar structure of RFEs to enhance their energy-storage performance[21–23]. Specially, the high-entropy strategy with introducing a minimum of five distinct ions in a solid solution has recently demonstrated as an effective method[24–27]. This approach enables the flexible design of different polarization configurations, such as polymorphic symmetry clusters, nanosized domains, multiple local distortions, by introducing various ions with different ionic radii,

[1]Beijing Advanced Innovation Center for Materials Genome Engineering, Department of Physical Chemistry, University of Science and Technology Beijing, Beijing 100083, China. [2]School of Materials Science and Engineering, Nanjing University of Science and Technology, Nanjing, Jiangsu 210094, China. [3]College of Materials Science and Engineering, Guilin University of Technology, Guilin 541004, China. [4]These authors contributed equally: Xingcheng Wang, Ji Zhang, Xingshuai Ma. ✉e-mail: huiliu@ustb.edu.cn; junchen@ustb.edu.cn

valence state, and ferroelectric activity[28–32]. Additionally, the high chemical disorder and inhibition of grain coarsening in high-entropy systems lead to grain refining, thereby enhancing $E_B$[33,34]. For example, a high $W_{rec}$ of 10.1 J cm$^{-3}$ and $\eta$ of 91% was achieved in (K,Na)TiO$_3$ (KNN)-based high-entropy bulk ceramic by delaying the polarization saturation[35]. The highest reported $W_{rec}$ of 13.8 J cm$^{-3}$ with $\eta$ of 82.4% was achieved in (Bi$_{0.5}$Na$_{0.5}$)TiO$_3$ (BNT)-based high-entropy bulk ceramic by increasing configurational entropy ($\Delta S$)[36]. Furthermore, ultra-high $W_{rec}$ was implemented in BaTiO$_3$ (BT)-based high-entropy multilayer ceramic capacitors (MLCCs) and Bi$_4$Ti$_3$O$_{12}$-based high-entropy thin films by increasing local compositional inhomogeneity and grain refinement[6,37]. Given that the compositional space of high-entropy RFEs is essentially infinite, the design of high-entropy materials has primarily focused on equimolar ratios and empirical trial-and-error methods[25,38]. Therefore, there is an urgent need for an effective approach to facilitate the screening of elements and compositions to explore high-performance high-entropy RFEs.

Recent advances in artificial intelligence and materials informatics, particularly machine learning (ML), have facilitated the accurate prediction of target properties for various material components within vast compositional spaces. ML has gained prominence in the field of FEs, where it has been utilized to extract and theoretically interpret fundamental descriptors, such as piezoelectric $d_{33}$, dielectric spectra, and accurately predicting new compositions or phase diagrams[39–43]. However, for energy-storage performance in bulk ceramics, the current research has focused on BT-based system, and the predicted low $W_{rec}$ at low $E$[44,45].

Here, a paradigm for discovering high-performance BNT-based high-entropy RFEs was introduced by leveraging ML. Utilizing a dataset comprising 121 reported BNT-based bulk ceramics, a model employing the random forest algorithm with a R$^2$ value of 0.84 was developed. Key descriptors were identified as the principle of screening elements, and four high-entropy compositions were predicted in the space of millions of compositions. Following by experimental validation, the special (Bi$_{0.36}$Na$_{0.34}$La$_{0.13}$Sr$_{0.17}$)(Ti$_{0.86}$Ta$_{0.01}$Mg$_{0.08}$Hf$_{0.05}$)O$_3$ high-entropy ($\Delta S$ ~ 1.8 R) near-linear RFEs with refined grains, weakly coupled polar clusters and low polarization anisotropy was obtained. Remarkably, a record-high $W_{rec}$ of 20.7 J cm$^{-3}$ with $\eta$ of 86% was achieved, outperforming the state-of-the-art Pb-free bulk ceramics. This study provides an effective method for designing high-entropy ceramics with superior energy-storage performance based on ML model building.

## Results and discussion
### Machine learning-driven high-entropy composition design
BNT-based solid-solution system is the most extensively and intensively studied perovskite-structure system for dielectric energy-storage owing to its relatively large polarization and excellent energy-storage properties[7,8]. Due to the more abundant data points of reported energy storage properties, the BNT-based system was selected for developing ML model. The small-scale dataset used for the ML model comprised 121 BNT-based energy-storage bulk ceramics from literatures, which includes chemical composition, $E_B$, and $W_{rec}$ (Supplementary Table 1). In the ML process, 16 A/B-site elements extracted from the dataset and 60 different descriptors were adopted (Fig. 1 a,b and Supplementary Table 2). These descriptors encompassed the physicochemical properties of the element, including those from the Mendeleev library, and Shannon's ionic radius. Furthermore, descriptors that have been shown to significantly impact ferroelectric properties, such as Vec-Z, octahedral factor (OF), Rdve and Rdce were also introduced[40,43–48]. However, these descriptors alone were insufficient to fit the regression model well. Therefore, the external feature of $E_B$ was also introduced to the model.

The construction of the ML model was divided into three main parts: feature engineering (Fig. 1c), model selection (Fig. 1d), and

regression fitting (Fig. 1e). Firstly, the redundant descriptors were eliminated to improve computational efficiency by analyzing the correlation between two descriptors. Highly correlated descriptors were grouped using the Pearson correlation coefficient plot. Within each group, a descriptor related to the target attribute was chosen to represent that group if the correlation coefficient was greater than 0.95 or less than −0.95. This filtering process resulted in 35 descriptors being retained for further analysis. The optimal number of descriptors was determined by employing the recursive feature elimination method, which is based on the R$^2$ and Mean Absolute Error (MAE) scores for various subsets of descriptors. The selection of the optimal number of descriptors occurred when the maximum R$^2$ value and the minimum MAE value were simultaneously achieved. As a result of the process, 11 descriptors were selected (Supplementary Fig. 2 and Table 3). Subsequently, five ML regression models were evaluated for predictive analysis of $W_{rec}$, including Random Forest (RF), Gradient Boosted Regressor (GBR), Decision Tree Regressor (DTR), Radial Basis Function Kernel Support Vector Regression (SVR.rbf), and Bayesian Ridge Regression (BR). Each model underwent a 10-fold cross-validation process, with hyperparameters adjusted to optimize performance[43–45,49]. The settings of the hyperparameters are detailed in Supplementary Tables 4–8. Based on the R$^2$ scoring function, RF presented the most favorable performance, with the highest R$^2$ values for the test set compared to the other models. Besides, the small difference in R$^2$ between the training and test sets indicates no overfitting for RF model. Finally, the 11 filtered descriptors and the RF model were employed as inputs for fitting. The results showed that the data points were distributed around the reference line with a small error margin, with a R$^2$ value of 0.84 and a MAE of 0.63 J cm$^{-3}$, indicating that the model exhibited good regression performance (Fig. 1e).

After randomly dividing the data into an 80% training set and a 20% test set, the random forest regression model was trained 25 times. The importance of the descriptors computed in each cycle was recorded, and finally, the average importance was computed and normalized (Fig. 1f). The analysis revealed that the three most crucial descriptors were the valence electron distance of the A-site, Allen electronegativity, and Ghosh electronegativity of the B-site. Analyzing the components of the dataset with high energy storage densities ($W_{rec}$ > 5 J cm$^{-3}$), it was found that the values of the three significant descriptors corresponding to them are in the smaller interval. This insight can guide the selection of elements. In the selection of A-site elements with smaller valence electron distances, it implies that the valence electrons of these elements are closer to the ionic core. This proximity increases the binding force on the electrons, reducing the likelihood of their escape under the influence of an external $E$, thereby minimizing charge accumulation. The B-site elements with lower electronegativity typically exhibit smaller average single-electron energies and atomic absolute radii of the valence shell layer electrons. This characteristic makes these elements more inclined to donate electrons during bond formation with oxygen, resulting in the creation of robust ionic bonds. These strong bonds enhance the structural stability as they are less susceptible to breakage under high electric fields, consequently increasing the $E_B$. Additionally, the smaller absolute radius of these atoms allows the B-site elements to be more tightly packed, forming a compact lattice structure, facilitating the $E_B$ and delaying the polarization saturation[50–52]. Figure 1g displays the values of important descriptors for various A-site and B-site elements, revealing that La, Ca, and Sr exhibit lower Rdve values at A-site. Notably, two significant descriptors were identified for the B-site elements. The weighting of the descriptors in conjunction with their significance magnitude in Fig. 1f served as a filtering criterion for the final selection of elements with lower values. Consequently, Hf, Mg, Zr, and Ta were identified as candidates for further evaluation based on this selection criterion. Interestingly, these identified A-site elements align with those recently reported in Bi(Mg$_{0.5}$Ti$_{0.5}$)O$_3$-based energy-

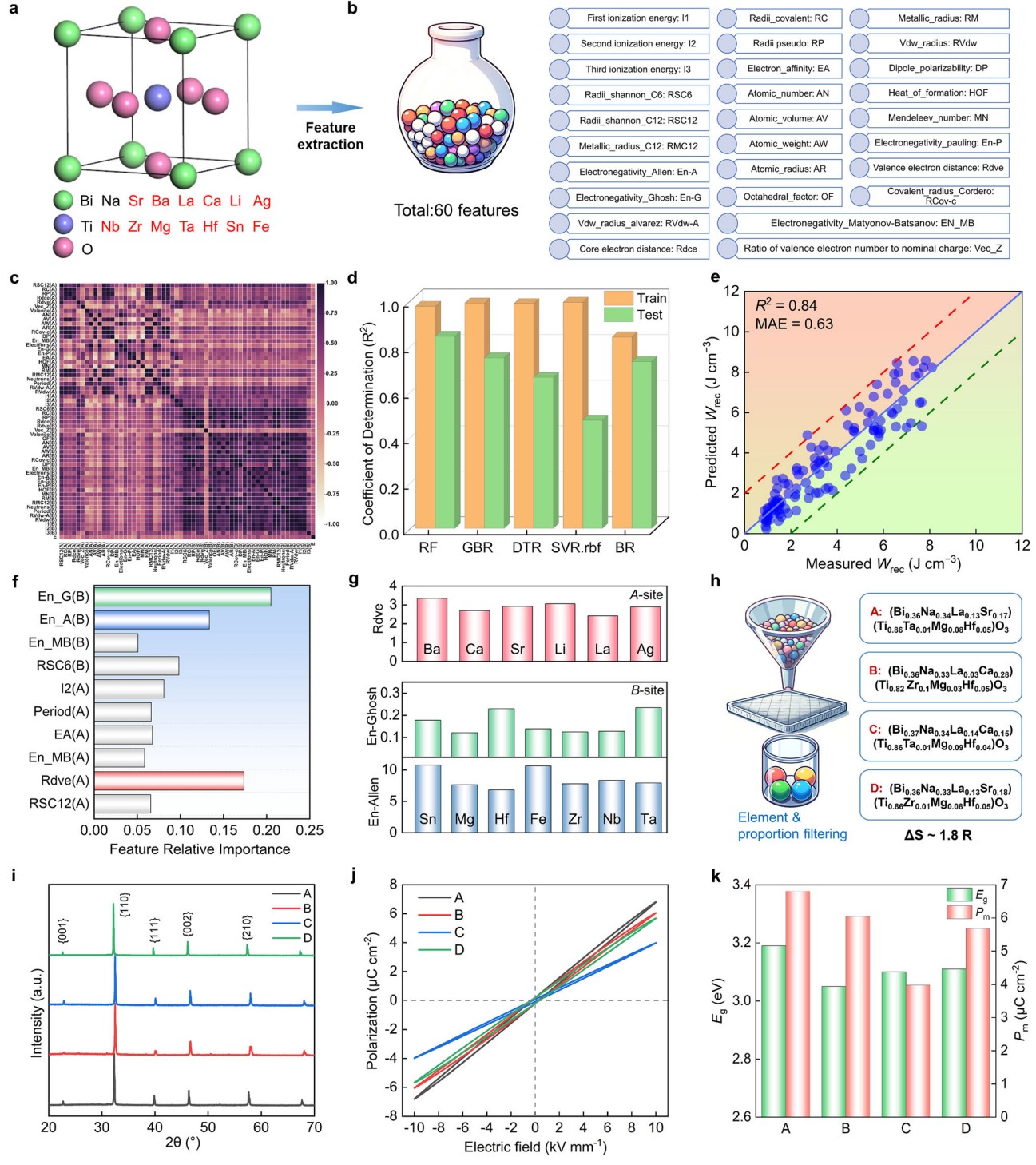

**Fig. 1 | Schematic illustrating the exploring of high energy-storage performance BNT-based high-entropy compositions through a combination of machine learning and experimental verification. a** $ABO_3$ perovskite structure combined with the elements during machine learning. **b** Representative descriptors (**c**) Pearson correlation matrix for 60 descriptors. **d** Model selection through cross-validation $R^2$ score including training and testing in different models. **e** Performance of using Random Forest (RF) model on the training data using the 11 descriptor sets. **f** Result of descriptors importance. **g** Comparison of important descriptors corresponding to different A-site and B-site elements. **h** Candidate compositions with the highest $W_{rec}$ obtained from element and proportion filtering. **i** XRD patterns, **j** P–E loops measured at low E, and **k** bandgap and $P_m$ of A, B, C, and D samples.

storage system[53]. Prior experiments have also reported that substituting Mg and Ta in BNT-based ceramics led to an increase in the bandgap and enhancement of the $E_B$[54]. This indicates that the developed ML model is effective in guiding the selection of elements based on their important descriptors. Based on the identified elements, the following ratio constraints were applied to the different doping

elements by substituting and combining these elements: $0 \leq x_{La} \leq 0.15$, $0 \leq x_{Sr} \leq 0.2$, $0 \leq x_{Ca} \leq 0.2$, $0 \leq x_{Ta} \leq 0.08$, $0 \leq x_{Zr} \leq 0.2$, $0 \leq x_{Mg} \leq 0.1$, $0 \leq x_{Hf} \leq 0.1$, with variations in steps of 0.01. These ratio constraints are largely in line with the ranges covered by the corresponding elements in the dataset, which enables a better match the established model as a way to achieve effective predictions. The optimized model predicted

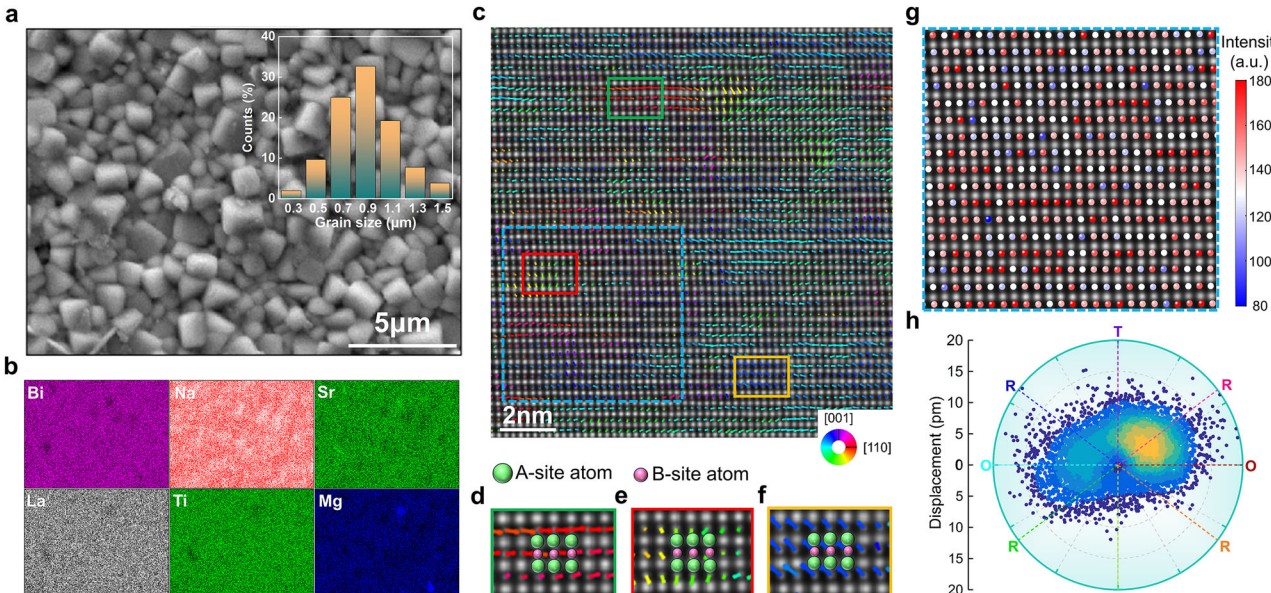

**Fig. 2 | Microstructures analysis of high-entropy sample A using electron microscopy techniques. a** Scanning electron microscopy (SEM) image displaying the surface morphology and grain size distribution. **b** Corresponding element distribution mapping. **c** High-angle annular dark-field scanning transmission electron microscopy (HAADF-STEM) image captured along the [110]$_c$ direction, accompanied by the mapping of B-site displacement vector. **d-f** Enlarged views of representative areas from **c**, illustrating the positions of the A-site and B-site atomic columns. **g** The B-site atomic intensities for the selected region. **h** Statistics of B-site polar displacement vectors.

all component points in the large unknown compositional space. Four specific high-entropy compositions with the highest predicted $W_{rec}$ were obtained, including A: $(Bi_{0.36}Na_{0.34}La_{0.13}Sr_{0.17})(Ti_{0.86}Ta_{0.01}Mg_{0.08}Hf_{0.05})O_3$, B: $(Bi_{0.36}Na_{0.33}La_{0.03}Ca_{0.28})$ $(Ti_{0.82}Zr_{0.1}Mg_{0.03}Hf_{0.05})O_3$, C: $(Bi_{0.37}Na_{0.34}La_{0.14}Ca_{0.15})(Ti_{0.86}Ta_{0.01}Mg_{0.09}Hf_{0.04})O_3$, and D: $(Bi_{0.36}Na_{0.33}La_{0.13}Sr_{0.18})(Ti_{0.86}Zr_{0.01}Mg_{0.08}Hf_{0.05})O_3$ (Fig. 1h). All of these compositions possess a $\Delta S$ value of about 1.8 R. Notably, the screened compositions suggest that a high-entropy composition with a moderate $\Delta S$ value would be more beneficial for achieving high energy-storage performance, which is consistent with recent experimental study[6].

To further determine the optimal composition among these four high-entropy compositions, ceramic samples were prepared by conventional solid-phase methods. Basic and facile experiments, including X-ray diffraction, P-E loops at low E, bandgap measurement, were conducted to further screen the compositions. These four samples exhibit pure perovskite structures (Fig. 1i). Interestingly, these samples present RFE behaviors with near-linear P-E loops and small hysteresis ($H < 10\%$) at low electric field (Fig. 1j), confirming the high reliability of the developed ML model. This also suggests that combining La, Sr, Mg, and Hf ions with the high-entropy strategy results in a weak coupled polar nanoregions (PNRs). It is well known that a large bandgap ($E_g$) usually corresponds to a high $E_B$ of dielectrics[55]. Accordingly, the $E_g$ values of these samples were measured (Fig. 1k and Supplementary Fig. 3), revealing significantly higher $E_g$ values compared to the energy-storage ceramics that have been reported so far[15,21,55,56]. Based on the combination of $E_g$ and $P_m$ values, sample A was selected, as it exhibited the largest maximum polarization ($P_m$) value of 6.8 μC cm$^{-2}$ ($E = 10$ kV mm$^{-1}$), as well as the highest $E_g$ of 3.18 eV among the four compositions. These properties are advantageous for achieving high-capacitive energy-storage, hence the high-entropy sample A was chosen to further explore its structure and energy storage characteristics.

## Local polar structure

The microstructures of high-entropy sample A were analyzed (Fig. 2). Sample A was sintered well, with fine and homogeneous grains of about 0.8 μm (Fig. 2a), which representing the smallest grain size compared to samples B, C, and D (Supplementary Fig. 4). The elements in the sample present uniform distribution, forming a single-phase high-entropy solid solution (Fig. 2b). The fine grain size in this high-entropy ceramic would be a result of the interactions among multiple elements, which increase the complexity of grain growth by causing diffusion inhomogeneity that hinders the movement of grain boundaries. The small average grain size indicates high fraction of grain boundaries within the same range, which is beneficial for achieving a high $E_B$[33,57].

In order to explore the polarization configuration of sample A, atomic-resolution HAADF images were recorded along the [110]$_c$ (Fig. 2c-f). The positions of the A-site (exhibiting stronger intensity contrast) and B-site (exhibiting weaker intensity contrast) atomic columns were analyzed by fitting a two-dimensional Gaussian function. This enabled the exploration of atomic displacement vectors, which were measured as the deviation of B-site atoms from the centers of their two nearest neighboring A-site atoms and represented as arrows. Note that the image taken from the [110]$_c$ direction can offer the advantage of better differentiating between the orthorhombic (O) and rhombohedral (R) symmetry distortion of the displacement[56,58]. Figure 2c-f shows the B-site displacement mapping, with RGB color conveying the direction and the arrow's length representing the atomic displacement amplitude. Clusters of coherent island-like displacement vectors with sizes ranging from 1–3 nm were detected. The coexistence of R, tetragonal (T), and O multi-symmetric clusters establishes a strong disorder. Moreover, the polar clusters are interconnected by lower symmetry clusters rather than being isolated by nonpolar matrices. Supplementary Fig. 5 comprehensively presents the magnitude and angle mappings of the atomic displacement vectors, with the largest estimated displacement, approximately 21 pm, attributed to the high content of strongly polarizable Bi and Na ions[15,21,56]. The occupation of A/B-sites by ions with varying radii, valence states, and ferroelectric activities results in high conformational entropy, a uniform distribution of components, and multiple intermediate displacement vector directions (Fig. 2g,h). This reduces the polarization anisotropy and polarization reorientation energy barrier, ultimately contributing to

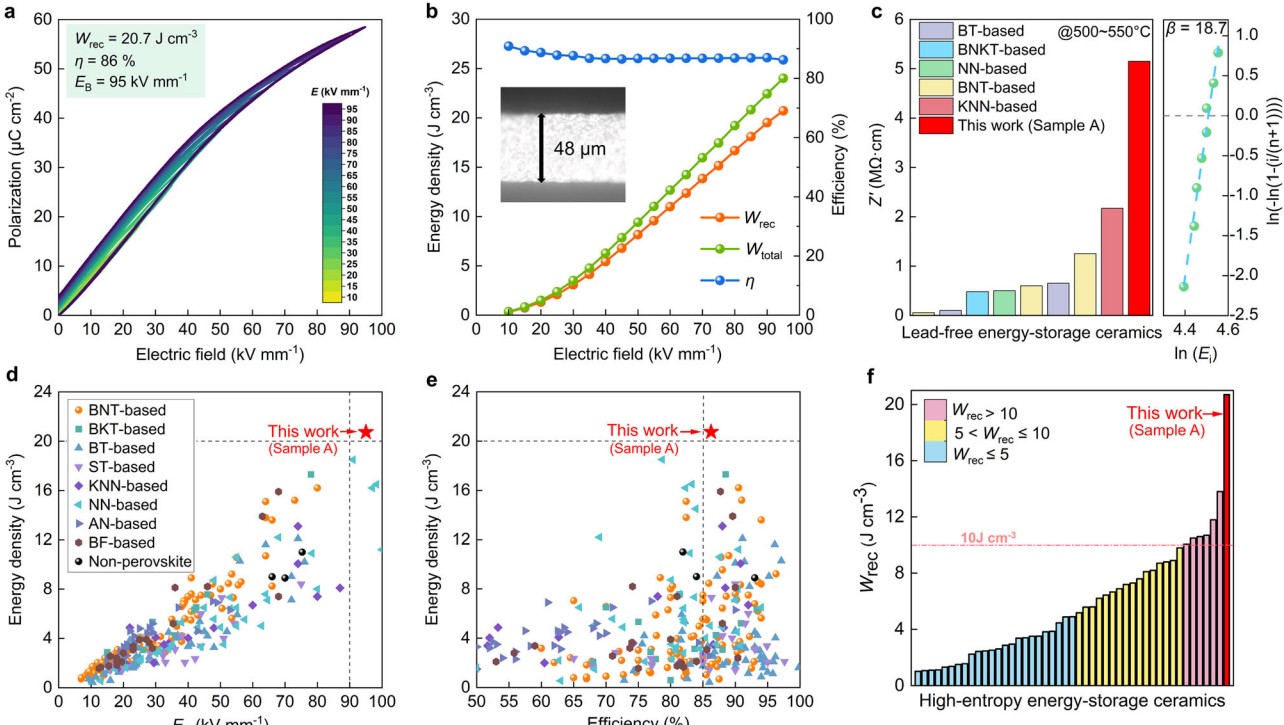

**Fig. 3 | Energy-storage capability of high-entropy sample A, measured using bulk ceramics with a thickness of about 50 μm and electrodes with an area of about 0.8 mm².** **a** Unipolar $P-E$ loops measured under different $E$. **b** Energy density and efficiency with respect to the applied $E$. The inset indicates the cross-sectional image of a bulk ceramic used for performance measurements. **c** Comparison of total electrical resistivity at around 500 - 550 °C and the Weibull distribution of the $E_B$ for sample A. Comparison of **d** $E_B$ and $W_{rec}$, **e** $W_{rec}$ and $\eta$ for lead-free bulk ceramics. **f** Comparison of $W_{rec}$ values for sample A with high-entropy engineered energy-storage bulk ceramics. Details about the properties and relevant references of the data points in the figures c-f can be found in Supplementary Tables 9–11.

the rapid polarization response to external fields and leading to reduced hysteresis[1,2,59].] This polar configuration in the high-entropy relaxor state can be reflected from the highly broadened dielectric maximum and large diffuseness parameter $\gamma$ of 2.2 observed in the dielectric spectrum (Supplementary Fig. 6). Hence, given the favorable polar structure in the high-entropy near-linear RFEs of sample A, it is expected to have excellent energy-storage performance.

**Energy-storage properties in bulk ceramic capacitors**

The energy-storage performance of the high-entropy sample A obtained through ML exploration was evaluated. Encouragingly, an excellent energy-storage performance were achieved (Fig. 3). Slim and near-linear $P-E$ loops were observed until to $E_B$. Particularly, a large $P_m$ of 58.5 μC cm$^{-2}$ and a negligibly $P_r$ of 3.8 μC cm$^{-2}$ were obtained at an ultrahigh $E_B$ of 95 kV mm$^{-1}$, resulting in a giant $W_{rec}$ of 20.7 J cm$^{-3}$ and high $\eta$ of 86%. The high $P_m$ and $E_B$, combined with the low $P_r$, contribute significantly to the excellent energy-storage performance. The $E_B$ obtained in high-entropy sample A is notably higher than the previously highest reported value in BNT-based systems in 80 kV mm$^{-1}$, and is comparable to the highest values observed in NN-based systems[57,60]. The observed ultrahigh $E_B$ can be attributed to several factors: 1) A refined grain size of 0.8 μm, as smaller grain size generally implies higher breakdown field strength[60]. The breakdown strength of ceramics increases with decreasing grain size, as small grain sizes increase the density of grain boundaries, and the hole charge layer accumulates at the grain boundaries, forming a barrier and leading to high resistivity[61]; 2) The ultrahigh Vickers hardness indicates good mechanical property that will enhance the electromechanical breakdown resistance (Supplementary Fig. 7); 3) the introduction of ion-type elements of Sr, Ta, Mg, and Hf leads to an increase in the $E_g$ compared

to BNT ceramic (Supplementary Fig. 3); 4) Sample A exhibits a remarkable high resistivity compared with other systems (Fig. 3c and Supplementary Fig. 8). All these factors lead to a low possibility of electric breakdown and high $E_B$.

The $W_{rec}$ and $\eta$ values were calculated based on the unipolar $P-E$ loops, as shown in Fig. 3b. The $\eta$ remains at a high level above 86% under applied different $E$. Remarkably, the achieved energy-storage performance of high-entropy sample A, with a $W_{rec}$ of 20.7 J cm$^{-3}$ and high $\eta$ of 86%, outperforms other bulk ceramics reported to date (Fig. 3 d-f)[7,15–18,24,36–38,56–59]. For examples, this performance is superior to the highest $W_{rec}$ value of 16.2 J cm$^{-3}$ in BNT-based bulk ceramics[57], the highest $W_{rec}$ values in bulk RFE ceramics (18.5 J cm$^{-3}$, 79%)[60], as well as the highest $W_{rec}$ value of 13.8 J cm$^{-3}$, $\eta$ of 82.4% among all the high-entropy engineered bulk RFE ceramics[36]. Additionally, the achieved performance is comparable the best MLCCs, such as the textured BNT–SBT (21.5 J cm$^{-3}$, 80%), high-entropy BT (20.8 J cm$^{-3}$, 97.5%)[2,5]. Even at 95 kV mm$^{-1}$, no sign of polarization saturation was observed, indicating that higher energy-storage could be achieved in MLCCs. Notably, the $W_{rec}$ value for sample A is significantly larger than the predicted value. This difference occurs because the $W_{rec}$ value is closely related to the $E_B$, which cannot be accurately predicted. The assumed value of the $E_B$ for prediction is 60 kV mm$^{-1}$, while the actual $E_B$ of sample A is much higher than this value. However, the experimental results still demonstrate that machine learning can be effective in screening components. Notably, the energy-storage performance of samples B, C and D was also evaluated (Supplementary Fig. 9). As anticipated, the sample A exhibits the best energy-storage performance among the four compositions, and is consistent well with the results from the basic and facile experiments. It indicates that the compositions predicted by ML can be further refined effectively.

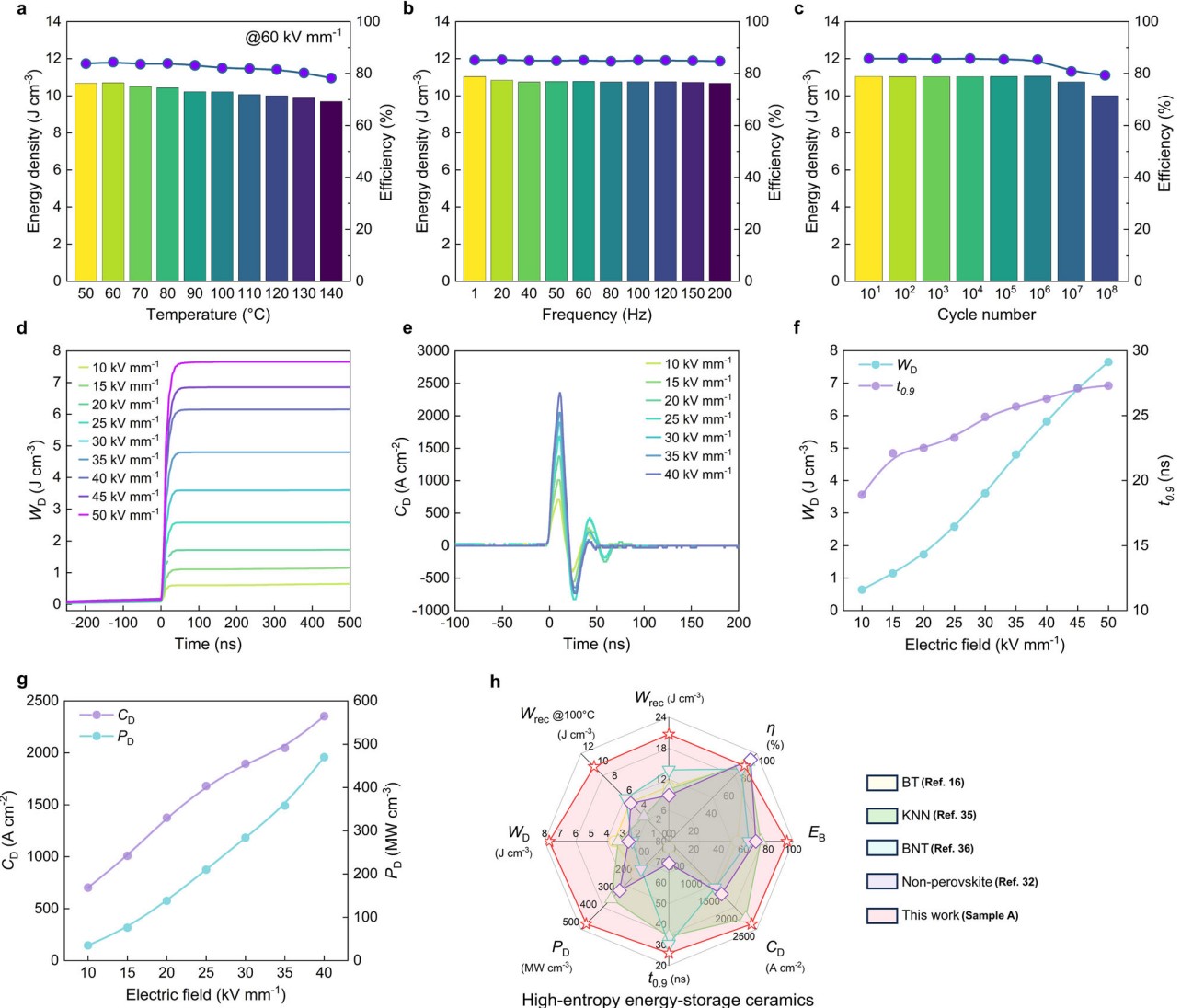

**Fig. 4 | Evaluation of the stability and charge/discharge energy-storage performance of high-entropy sample. a** Energy density and efficiency as a function of **a** temperature, **b** frequency, and **c** cycle number under 60 kV mm⁻¹. **d** Discharge energy density curves for an overdamped system ($R = 51\,\Omega$) and **e** for an underdamped system ($R = 51\,\Omega$). **f** $W_D$ versus $t_{0.9}$ and **g** $C_D$ versus $P_D$ as a function of $E$. **h** Comprehensive comparison of the energy-storage characteristics.

## Charge discharge performance and stability

Since dielectric ceramic capacitors are often required to operate in severe environments, the stability of high-entropy sample A was evaluated to determine its practical applicability. The unipolar P-E loops were performed at approximately 60% of the $E_B$, which was under 60 kVmm⁻¹ (Supplementary Fig. 10). As shown in Fig. 4a-c, a high $W_{rec}$ value around 10 J cm⁻³ can all be maintained under different severe environments. Under a temperature range of RT-140 °C, the energy storage density of about 10 J cm⁻³ and energy storage efficiency exhibit slight fluctuations, with variations within 8% and 7%, respectively (Fig. 4a), indicating that it can be used at high temperatures. Sample A demonstrates excellent frequency stability of energy storage density and energy storage efficiency across a wide frequency range of 1–200 Hz, with almost negligible variation ($W_{rec} \sim 10.8 \pm 0.2$ J cm⁻³, $\Delta\eta < 0.7\%$) (Fig. 4b). The energy-storage density and efficiency of sample A present essentially stable in performance from 10¹-10⁶ cycles ($W_{rec} \sim 11.0 \pm 0.05$ J cm⁻³, $\Delta\eta < 0.5\%$) (Fig. 4c), demonstrating excellent fatigue resistance and very high practical service life. To assess the application potential, the discharge performance of sample A was evaluated (Fig. 4d-g). The $W_D$ reaches its maximum value within 45 ns, progressively increasing with the electric field. It only takes $t_{0.9} = 27.3$ ns of ultrafast time at 50 kV mm⁻¹ to discharge 90% of the

saturated storage energy density, leading to an ultrahigh discharge energy density ($W_D = 7.7$ J cm⁻³). Moreover, the underdamped ($R = 51\,\Omega$) discharge current density curve in Fig. 4e exhibits regular oscillating waveforms with excellent current density ($C_D = 2353$ A cm⁻²) and ultrahigh power density ($P_D = 470$ MW cm⁻³) at 40 kV mm⁻¹, indicating excellent discharge capability. Compared to BNT-based ceramics with the highest $W_{rec}$ reported so far ($W_D = 5.2$ J cm⁻³, $P_D = 357$ MW cm⁻³). Further, a comparison of the comprehensive properties of sample A and representative Pb-free high-entropy bulk ceramics is shown in Fig. 4h. It covers a wide region of the radar diagram, demonstrating significant improvement and distinct advantages[16,32,35,36]. Given this comprehensive performance, sample A demonstrate great potential for application in advanced energy-storage capacitors[57].

In summary, it is demonstrated that a combination of machine learning prediction and screening, as well as basic experiments can be effective in exploring high energy-storage performance high-entropy RFEs. Through this strategy, an ultrahigh $W_{rec}$ of 20.7 J cm⁻³ with high $\eta$ of 86% under an ultrahigh $E_B$ of 95 kV mm⁻¹ has been achieved in a BNT-based high-entropy composition. Based on limited reported experimental data, A random forest regression model with key descriptors has been developed to guide the selection of elements, and efficiently

screen high-entropy compositions from in the vast space of millions of compositions. As an example, the machine learning model has successfully predicted four $(Bi_{0.5}Na_{0.5})TiO_3$-based high-entropy near-linear RFEs. Subsequently, basic experiments have been conducted to identify an excellent relaxor, characterized by fine grains, weak coupling and small-sized polar clusters. The synergistic high-entropy effect results in a record-high energy-storage performance. In addition, this developed material exhibits excellent thermal, frequency and cycling stability, an ultrahigh $W_D$ of 7.7 J cm$^{-3}$ and a large $P_D$ of 470 MW cm$^{-3}$. Our results show the data-driven model for efficiently exploring high-performance high-entropy bulk relaxor ceramics, demonstrating the potential of machine learning in designing complex high-entropy ferroelectric materials.

## Methods
See details of the methods part in the Supplementary Information files.

## Data availability
Relevant data supporting the key findings of this study are available within the paper and the supplementary information file. Source data are provided with this paper.

## Code availability
The related codes in this work are available in Code Ocean at: https://codeocean.com/capsule/4347581/tree/v2.

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

## Acknowledgements

This work was supported by the Key research and development Program of Ministry of Science and Technology of China (No. 2022YFB3204000, J.C.), the Outstanding Young Scientists Program of Beijing Higher Education Institutions (JWZQ20240101015, J.C.), and the National Natural Science Foundation of China (Nos. 22235002, J.C. and 22471013, H.L.).

## Author contributions

J.C. and H.L. conceived this study. X.C.W. performed the experiments under the supervision of H.L. and J.C.; X.C.W. and J.Z. conducted the energy-storage performance measurements. L.J.L and H.J.L. performed dielectric and impedance tests. X.C.W., H.L. and X.S.M. established the machine learning model. X.C.W. and H.L. carried out the STEM analysis. X.C.W. wrote the first draft of the manuscript. H.L. and J.C. revised the paper. All authors discussed the results and revised the manuscript.

## Competing interests

The authors declare no competing interests.
