## [Transparent Peer Review file · Nature Communications]

Machine learning assisted composition design of high-entropy Pb-free relaxors with giant energy-storage

Corresponding Author: Professor Jun Chen

Version 0:

Reviewer comments:

Reviewer #1

(Remarks to the Author)

The present manuscript by Wang et al., demonstrates ML assisted material development for superior energy storage. Based on the inputs obtained from literatures, the authors developed four BNT based ceramic showing high properties. Present manuscript is interesting but contains several shortcomings which need to be addressed. My comments are following:

1. Fig. 1c is hardly readable
2. It would be nice to specify which 35 features were selected after the checking for strong correlations, and which final 11 features remained after the recursive feature elimination (can be listed in the supplementary).
3. Readers would appreciate seeing in the manuscript MAE obtained for the test data.
4. For the best regressor, RF, it would be nice to list its optimized hyperparameters in the supplementary.
5. It would be nice to hear the author's comments on why certain ratio constraints were chosen.
6. It is good to mention the exact sample name (Among A,B , C, D) in the insets of the figures (Showing the term: 'This work').
7. Why bipolar PE loops are shown at an electric field an order of magnitude lower than that shown in case of unipolar loop?
8. The energy storage properties obtained using ML is just an incremental progress compare to the existing and no new mechanism/phenomenon is underlying to explain the properties. There are several other reports showing capacitive energy storage density reached $\sim 100 \text{ J/cm}^3$. So the obtained value here is not very exciting.
9. It is big surprise that breakdown strength in ceramic is so high.
10. Authors did not mentioned the references in the supplementary and in the figure captions.

(Remarks on code availability)

Reviewer #2

(Remarks to the Author)

In this study, the authors utilized machine learning to forecast and filter the elements and chemical compositions of high-entropy Pb-free relaxors, achieving an unprecedented high energy-storage density of 20.7 Jcm^{-3} with a high efficiency of 86%. However, several issues need to be addressed before the research can be considered for publication:

1. The dataset of 121 BNT-based energy-storage bulk ceramics appears insufficient for the ML model, as machine learning typically requires large-scale training data for reliable results. The authors should provide additional details and clarify the benefits of using a small-scale training dataset to better demonstrate the efficacy of their method.
2. The authors tested five ML regression models for predicting W_{rec} but did not provide data to support the choice of the random forest regression model.
3. The authors assert that they developed a random forest regression model for this study. What novel features does this model have compared to existing models?
4. The optimization method used in this study is not described. Please provide a detailed explanation, and there is no description of the machine learning methodology in the methods section.

5. The rationale for selecting BNT as the base system and specific elements for doping is not explained.
6. The study applied different ratio constraints for various doping elements (e.g., $0 \leq x_{La} \leq 0.15$, $0 \leq x_{Sr} \leq 0.2$, $0 \leq x_{Ca} \leq 0.2$, $0 \leq x_{Ta} \leq 0.08$, $0 \leq x_{Zr} \leq 0.2$, $0 \leq x_{Mg} \leq 0.1$, $0 \leq x_{Hf} \leq 0.1$), but the reasons for these specific constraints are unclear.
7. Grain size significantly influences ferroelectric properties and energy storage, yet there is no microstructure analysis of samples B, C, and D to confirm similar grain sizes among all samples.
8. The authors optimized samples A, B, C, and D by comparing the largest maximum polarization value at 10 kV/mm and bandgap, but why not directly compare the energy storage capacity and breakdown electric fields of these samples?
9. The claim that "smaller grain size generally implies higher breakdown field strength" needs supporting evidence.
10. The horizontal axis in Fig. 4c lacks a label.

(Remarks on code availability)

Version 1:

Reviewer comments:

Reviewer #1

(Remarks to the Author)

I am happy that authors has put great efforts for the improvement of the manuscript. However, unit of dielectric breakdown strength reported in supplementary table 9 is kV/cm while the unit used in elsewhere is kV/mm. Here is the confusion. If I believe that unit kV/mm shown in the figure 3a is correct, then break down strength of the reported bulk ceramic to be around 950 kV/cm which is hard to imagine in the bulk ceramic.

Author should clarify it and be consistent throughout the manuscript.

(Remarks on code availability)

Reviewer #2

(Remarks to the Author)

After evaluating the changes made by the authors in response to the reviewers' comments, this referee finds that most aspects of the study have been addressed. The authors have tried to address the previous comments and questions that are crucial for the readers to make deeper understanding that they claim to demonstrate. The work is suited for publication in a journal with the scope of Nature Communications.

(Remarks on code availability)

Version 2:

Reviewer comments:

Reviewer #1

(Remarks to the Author)

Authors have corrected the error and revised the manuscript. Now the article can be considered for publication in Nature Communications.

(Remarks on code availability)

Dear reviewers,

Thank you so much for spending your time reading our manuscript and providing insightful comments. In the following, we present our response to your comments, point by point. Based on your comments and suggestions, we have further improved the manuscript. We hope that these changes adequately address the concerns raised. We have highlighted all the changes within the manuscript in the blue-colored text.

Reviewer #1 (Remarks to the Author):

The present manuscript by Wang et al., demonstrates ML assisted material development for superior energy storage. Based the inputs obtains from literatures, the authors developed four BNT based ceramic showing high properties. Present manuscript is interesting but contains several shortcomings which need to be addressed. My comments are following:

Reply: We thank reviewer for the positive opinion about our work. We have carefully addressed the concerns raised by the reviewer. All changes have been highlighted in the revised manuscript.

Comment 1: Fig. 1c is hardly readable

Reply 1: Thanks for comment. In order to plot the Fig. 1c more clearly, we have abbreviated Octahedral_factor to OF to improve the image layout. Accordingly, we have updated the Figure 1b, Fig. 1c and the Supplementary Table 2.

Revision: We have revised Fig. 1b, Fig. 1c on page 4 of the manuscript and the corresponding parts of Supplementary Table 2.

Comment 2: It would be nice to specify which 35 features were selected after the checking for strong correlations, and which final 11 features remained after the recursive feature elimination (can be listed in the supplementary).

Reply 2: Thanks for the comments. We have added the remaining features after strong correlation check and recursive feature elimination in the Supplementary Material. The added Supplementary Table 3 is pasted here for reference.

Supplementary Table 3. The strong correlation check and recursive feature elimination results in remaining descriptors.

	After the checking for strong correlations (35 features)			After the recursive feature elimination (11 features)
Features	RSC12(A)	En-P(A)	Vec_Z(B)	RSC12(A)
	RC(A)	EA(A)	Valence(B)	Rdve(A)
	Rdce(A)	HOF(A)	AN(B)	En_MB(A)
	Rdve(A)	MN(A)	DP(B)	EA(A)
	Vec_Z(A)	Period(A)	En_MB(B)	Period(A)
	Valence(A)	I1(A)	En-A(B)	I2(A)
	AN(A)	I2(A)	En-G(B)	RSC6(B)
	AV(A)	I3(A)	MN(B)	En_MB(B)
	AR(A)	RSC6(B)	Period(B)	En-A(B)
	DP(A)	RP(B)	I3(B)	En-G(B)
	En_MB(A)	Rdce(B)	Electric field	Electric field
	En-G(A)	Rdve(B)		

Revision: We have added all these in the Supplementary Table 3, and discussions in the manuscript.

Page 5: “The selection of the optimal number of descriptors occurred when the maximum R^2 value and the minimum MAE value were simultaneously achieved. As a result of the process, 11 descriptors were selected (Supplementary Fig. 2 and Table 3).”

Comment 3: Readers would appreciate seeing in the manuscript MAE obtained for the test data.

Reply 3: Thanks for the comment. In the present study, the MAE of the final random forest regression model is 0.63 J cm^{-3} , indicating that the model has a small error and is well fitted to effectively predict the new components. We have added the mean absolute error (MAE) for the test data of the regression model in Fig. 1e and added description in the corresponding section of the manuscript.

Revision: We have revised Fig. 1e on page 4 and the corresponding descriptions on page 6 of the revised manuscript.

Page 6: “The results showed that the data points were distributed around the reference line with a small error margin, with an R^2 value of 0.84 and an MAE of 0.63 J cm^{-3} , indicating that the model exhibited good regression performance (Fig. 1e).”

Comment 4: For the best regressor, RF, it would be nice to list its optimized hyperparameters in the supplementary.

Reply 4: Thanks for the comment. In searching for suitable model hyperparameters, we selected the optimal hyperparameter combination by traversing the given hyperparameter space through a grid search, training each set of possible hyperparameter combinations, and evaluating their performance on the validation set. We have listed the five models considered in the manuscript (including random forests), the hyperparameters finally chosen, and the corresponding training results in the Supplementary Tables 4-8. The added Supplementary Tables 4-8 are pasted here for your reference.

Supplementary Table 4. Random forest training results and the used hyperparameters.

	n_estimators	max_depth	random_state	min_samples_split	min_samples_leaf	R^2 (test)	R^2 (train)
RF	200	8	None	2	1	0.84	0.97

Supplementary Table 5. Gradient Boosted Regressor training results and the used hyperparameters.

	loss	n_estimators	max_depth	learning_rate	min_samples_split	R^2 (test)	R^2 (train)
GBR	squared_error	200	8	0.0115	2	0.74	0.98

Supplementary Table 6. Decision Tree Regressor training results and the used hyperparameters.

	max_depth	random_state	min_samples_split	R^2 (test)	R^2 (train)
DTR	8	None	4	0.66	0.98

Supplementary Table 7. Radial Basis Function Kernel Support Vector Regression training results and the used hyperparameters.

	kernel	C	gamma	R^2 (test)	R^2 (train)
SVR.rbf	rbf	1000	0.0075	0.47	0.99

Supplementary Table 8. Bayesian Ridge Regression training results and the used hyperparameters.

	tol	fit_intercept	max_iter	compute_score	R^2 (test)	R^2 (train)
BR	1e-5	False	300	True	0.73	0.84

Revision: We have added all these in the revised Supplementary Information.

Comment 5: It would be nice to hear the author's comments on why certain ratio constraints were chosen.

Reply 5: Thank you for your comments. We would like to point that for the dopant elements, such as La, Ta, Zr, Mg and Hf, the ratio constraints set are in line with the range covered in the Group 121 dataset. It allows to match as much as possible the modelling established in order to make valid predictions. For Ca and Sr, it has been shown in the literature that the high content will lead to a significant decrease in the polarization value. Therefore, we appropriately narrowed the range of their ratio constraints, which are of course within the range covered by the dataset. In this way, the predictions are optimized as much as possible while maximally fitting the model range. We also provide explanations in the corresponding sections of the manuscript.

Revision: We have added discussions of elemental ratio constraints to the manuscript.

Page 7: Based on the identified elements, the following ratio constraints were applied to the different doping elements by substituting and combining these elements: $0 \leq x_{La} \leq 0.15$, $0 \leq x_{Sr} \leq 0.2$, $0 \leq x_{Ca} \leq 0.2$, $0 \leq x_{Ta} \leq 0.08$, $0 \leq x_{Zr} \leq 0.2$, $0 \leq x_{Mg} \leq 0.1$, $0 \leq x_{Hf} \leq 0.1$, with variations in steps of 0.01. The ratio constraints are largely in line with the ranges covered by the corresponding elements in the dataset, which enables a better match the established model as a way to achieve effective predictions.

Comment 6: It is good to mention the exact sample name (Among A, B, C, D) in the insets of the figures (Showing the term: 'This work').

Reply 6: Thanks for the comment. In the manuscript, we have added the words "Sample A" to highlight this.

Revision: We have revised Figures 3c-f on page 10 and Figure 4h on page 13 of the manuscript.

Comment 7: Why bipolar PE loops are shown at an electric field an order of magnitude lower than that shown in case of unipolar loop?

Reply 7: Thanks for the comment. We would like to point that the bipolar *P-E* loops shown in Fig. 1j were measured using bulk ceramics samples with a thickness of about 0.5 mm and a diameter of 8 mm. The measurement of *P-E* loops at low electric field is very easy and convenient. Therefore, we the easy obtained *P-E* loops at low electric fields were used to provide some indication of the energy storage properties of the samples at high electric fields. It can be used to screen the samples very effectively.

In contrast, when evaluating the energy storage properties of samples by measuring P - E loops at high electric fields, it is usually necessary to cost a lot of time to polish the samples down to about 50 μm . Therefore, we used the easy obtained P - E loops at low electric fields to provide some indication of the energy storage properties of the samples at high electric fields. It is worth mentioning that the energy storage test of ferroelectric thin films is different compared with bulk ceramics.

(2) On the other hand, in the revision, we also tested the unipolar P - E loops of samples B, C, and D under high electric fields. The results demonstrate that sample A has the highest P_m at low electric field and still has the highest value at high electric field; Additionally, sample A, with the highest band gap, had the highest E_B and the best energy storage performance.

Revision: We have added some explanation in the revised manuscript.

Page 7: To further determine the optimal composition among these four high-entropy compositions, ceramic samples were prepared by conventional solid-phase methods. Basic and facile experiments, including X-ray diffraction, P - E loops at low E , bandgap measurement, were conducted to further screen the compositions.

Page 13: Notably, the energy-storage performance of samples B, C and D was also evaluated (Supplementary Fig. 9). As anticipated, the sample A exhibits the best energy-storage performance among the four compositions, and is consistent well with the results from the basic and facile experiments. It indicates that the compositions predicted by ML can be further refined effectively.

Comment 8: The energy storage properties obtained using ML is just an incremental progress compare to the existing and no new mechanism/phenomenon is underlying to explain the properties. There are several other reports showing capacitive energy storage density reached $\sim 100 \text{ J/cm}^3$. So the obtained value here is not very exciting.

Reply 8: Thanks for the comments. We agree with the reviewer's perspective. Because relying solely on experimental trial-and-error approaches to discover high-performance high-entropy compositions is an arduous task. Our work demonstrates that machine learning is a highly effective tool for accelerating the search for novel high-energy-storage performance high-entropy materials. In the present study, the random forest model effectively captures the nonlinear relationships between features, which is one of the reasons for the remarkable enhancement in the performance of the new compositions.

We also noted the several nice and excellent works about leveraging machine learning to

explore high-energy-density in thin films, such as 156 J cm^{-3} in BMT-based thin films (*Nat. Commun.*, 15, 4940, 2024), 80 J cm^{-3} in BT-based thin films (*Nano Lett.*, 23, 4807, 2023). It should be noted that all of the excellent energy storage properties reported in these works were based on thin films, which have a thickness (500 nm) much smaller than that of bulk ceramics. Based on the current research progress, achieving energy storage density of around 100 J cm^{-3} is relatively easy in thin films.

In contrast, for energy-storage bulk ceramics, the energy storage density is typically about 10 J cm^{-3} as shown in Fig. 3d. The reported energy-storage density of 20.7 J cm^{-3} in the present study using machine learning is among the highest values of reported for all bulk ceramics.

Comment 9: It is big surprise that breakdown strength in ceramic is so high.

Reply 9: Thank you for your comment. The high breakdown strength of Sample A can be attributed to several factors: (1) Sample A has a small grain size of $0.8 \mu\text{m}$, which increases the density of grain boundaries where a layer of hole charges builds up, forming a potential barrier and leading to a high resistivity (*Prog. Mater. Sci.* 102, 72–108 2019). (2) Sample A exhibits a substantially high resistivity compared to the other reported energy-storage systems according to the complex impedance spectra results (Fig. 3c). (3) Sample A has the highest bandgap about 3.2 eV among all the samples. Breakdown strength is usually proportional to the band gap (*Chem. Rev.* 121, 6124-6172, 2021). (4) The ultrahigh Vickers hardness of Sample A indicates good mechanical property that will enhance the electromechanical breakdown resistance (Supplementary Fig. 7). All these factors lead to a low probability of electrical breakdown and ultrahigh breakdown strength (Fig. 3c).

Revision: We have added some explanation in the revised manuscript.

Page 10: The observed ultrahigh E_B can be attributed to several factors: 1) A refined grain size of $0.8 \mu\text{m}$, as smaller grain size generally implies higher breakdown field strength.⁶⁰ The breakdown strength of ceramics increases with decreasing grain size, as small grain sizes increase the density of grain boundaries, and the hole charge layer accumulates at the grain boundaries, forming a barrier and leading to high resistivity;⁶¹ 2) The ultrahigh Vickers hardness indicates good mechanical property that will enhance the electromechanical breakdown resistance (Supplementary Fig. 7); 3) the introduction of ion-type elements of Sr, Ta, Mg, and Hf leads to an increase in the E_g compared to BNT ceramic (Supplementary Fig. 3); 4) Sample A exhibits a remarkable high resistivity compared with other systems (Fig. 3c and

Supplementary Fig. 8). All these factors lead to a low possibility of electric breakdown and high E_B .

Comment 10: Authors did not mentioned the references in the supplementary and in the figure captions.

Reply 10: Thanks for the comment. The major references are shown in the main text. Due to too many reported data points in the comparison of properties in the Fig. 3. We have listed all relevant references in the Supplementary material (Tables S9-S11). For the comparison of comprehensive properties in Fig. 4h, the reference is shown in Fig. 4h.

Reviewer #2 (Remarks to the Author):

In this study, the authors utilized machine learning to forecast and filter the elements and chemical compositions of high-entropy Pb-free relaxors, achieving an unprecedented high energy-storage density of 20.7 J cm^{-3} with a high efficiency of 86%. However, several issues need to be addressed before the research can be considered for publication:

Reply: We highly appreciate your positive overall comments. We have carefully addressed the concerns raised by the reviewer through additional charts and discussion. All changes have been highlighted in the revised manuscript.

Comment 1: The dataset of 121 BNT-based energy-storage bulk ceramics appears insufficient for the ML model, as machine learning typically requires large-scale training data for reliable results. The authors should provide additional details and clarify the benefits of using a small-scale training dataset to better demonstrate the efficacy of their method.

Reply: Thank you for the comments. As the reviewer stated machine learning usually requires large-scale training data to produce reliable results. However, sometimes too much data would also contain more noise, causing the model to try to ‘remember’ the details and fall into overfitting and make incorrect predictions. For smaller dataset, the model is more likely to look for the most significant trends in the data.

The dataset constructed in the present study was carefully selected to exclude samples with extreme values of energy storage density and samples involving infrequently used dopants to ensure that there are enough samples for each dopant, which improves the quality and representativeness of the dataset and contributes to the accuracy of the model training. In addition, we have plotted learning curves to better explain this, and relevant content has been added to the Supplementary Information. All are pasted here for your reference.

6. Dataset sample size reasonableness validation: In order to verify the effectiveness of a relatively small amount of data for machine learning, the learning curve of the optimized random forest model showing the training scores for different sample sizes in the training set is provided in Supplementary Figure 1. It can be seen that when the sample size in the training set exceeds 80, the validation set score reaches a high and stable level, and the model is not affected by overfitting or underfitting. This is a good indication that the model can learn effective information and make accurate predictions even with a relatively small amount of data.

Supplementary Fig. 1 Learning Curves for Random Forest regression model on the dataset.

Revision: We added them to the Methods section of the Supplementary Information and to Supplementary Fig. 1.

Comment 2: The authors tested five ML regression models for predicting W_{rec} but did not provide data to support the choice of the random forest regression model.

Reply 2: Thank for the comments. We have added a more detailed description to the model selection section of the manuscript to explain the reasons for choosing the random forest regression model, and have listed the hyperparameter settings for the different models and the corresponding R^2 scores in the revised Supplementary. All are pasted here for your reference.

Supplementary Table 4. Random forest training results and the used hyperparameters.

	n_estimators	max_depth	random_state	min_samples_split	min_samples_leaf	R^2 (test)	R^2 (train)
RF	200	8	None	2	1	0.84	0.97

Supplementary Table 5. Gradient Boosted Regressor training results and the used hyperparameters.

	loss	n_estimators	max_depth	learning_rate	min_samples_split	R^2 (test)	R^2 (train)
GBR	squared_error	200	8	0.0115	2	0.74	0.98

Supplementary Table 6. Decision Tree Regressor training results and the used hyperparameters.

	max_depth	random_state	min_samples_split	R^2 (test)	R^2 (train)
DTR	8	None	4	0.66	0.98

Supplementary Table 7. Radial Basis Function Kernel Support Vector Regression training results and the used hyperparameters.

	kernel	C	gamma	R^2 (test)	R^2 (train)
SVR.rbf	rbf	1000	0.0075	0.47	0.99

Supplementary Table 8. Bayesian Ridge Regression training results and the used hyperparameters.

	tol	fit_intercept	max_iter	compute_score	R^2 (test)	R^2 (train)
BR	1e-5	False	300	True	0.73	0.84

Revision: We have added all these in the Supplementary Tables 4-8, and discussions in the manuscript.

Page 6: Based on the R^2 scoring function, RF showed the most favourable performance, with the highest R^2 values for the test set and a small difference in R^2 between the training and test sets compared to the other models, indicating no overfitting.

Comment 3: The authors assert that they developed a random forest regression model for this study. What novel features does this model have compared to existing models?

Reply 3: Thanks for the comments. Indeed, there are a few publications that report the application of machine learning approaches for dielectric energy-storage. For example, in the recent work on BMT thin films for energy storage, it solved the problem of insufficient amount of data through generative learning by using data reconstruction and artificial neural networks (ANNs) to ultimately screen out superior high-entropy thin film components (*Nat. Commun.*, 15, 4940, 2024). Compared to it, the random forest regression model we used has new features, such as (1) It is learning directly from existing data and finding the relationship between input features and energy storage density by means of regression trees. Instead of assuming or generating new data as in generative learning, this approach models directly from existing data, thus reducing the dependence on the process of generating new data. (2) Random forest regression is usually more interpretable than generative learning methods. By ranking the importance of each feature, this helps us to clearly understand which descriptors have a significant impact on the predicted results, which helps to make sound material design decisions.

In previous work on machine learning prediction of energy storage ceramics, it used a classification model to find compounds belonging to the cross-region between the ferroelectric phase and the relaxation ferroelectric phase, and then built a support vector machine regression

model to predict and experimentally design these compounds (*Adv. Sci.* 6, 1901395, 2019). In contrast to it, the Random Forest regression approach is completely data-driven, extracting patterns directly from the data and predicting unknown compounds. Its advantage lies in its ability to automatically learn to complex non-linear relationships and to capture underlying patterns from data without the need for a priori physical assumptions.

Thus, Random Forest is a powerful method in integrated learning, based on the idea of decision trees, which improves the performance of the model through randomization and voting mechanisms. Each tree is independent during the training process, and by integrating multiple decision trees increases the diversity among different trees and reduces the bias of a single tree on a specific training set, which in turn can reduce the bias and variance of the model and improve the generalization ability of the model.

Comment 4: The optimization method used in this study is not described. Please provide a detailed explanation, and there is no description of the machine learning methodology in the methods section.

Reply 4: Thanks for the comments. In the Methods section of the revised Supplementary, we have added detailed descriptions of each step of the methodology and optimization process in machine learning and pasted here for your reference.

Machine learning:

1. Dataset and feature pool: The dataset of BNT-based ceramics investigated in this study was collected from the published literature (see Supplementary Table 1 for detailed data). We selected various descriptors in terms of structural and geometrical properties, electron and charge distribution properties, electronegativity, and nucleus properties (see Supplementary Table 2 for details) and generated feature values matrix for samples in the dataset by weighted summation.

2. Feature selection: Pearson correlation analysis was used to identify key features of the machine learning model. The Pearson correlation coefficient (r) is mathematically expressed as:

$$r = \frac{\sum_{i=1}^n (x_i - \bar{x})(y_i - \bar{y})}{\sqrt{\sum_{i=1}^n (x_i - \bar{x})^2 \sum_{i=1}^n (y_i - \bar{y})^2}}$$

Here, n denotes the number of features, x_i and y_i represent two distinct features, with \bar{x} and \bar{y} being their respective means. The higher the value of r , the more strongly the two features are linearly related. Setting 0.95 is a threshold for the correlation coefficient to determine if features are too similar to each other. If the correlation coefficient of two features is greater than or equal to this threshold, they are considered highly correlated. For the group of features with high correlation, one of the features is retained and the other related features are removed. The goal of this step is to reduce the dimensionality of the feature space while reducing the risk of overfitting the model. After screening by Pearson's correlation coefficient, the remaining features are usually those that have a strong correlation with the target variable (y) and a low correlation between them and other features. These features retain more useful information and help to improve the performance of the model. Subsequently, through the combination of recursive feature elimination and cross-validation, the best subset of features is identified by eliminating one feature at a time and iterating continuously, using 10-fold cross-validation to evaluate the model performance after each feature elimination.

3. Model selection: The performance of various machine learning regression algorithms, including Random Forest (RF), Gradient Boosted Regressor (GBR), Decision Tree Regressor (DTR), Radial Basis Function Kernel Support Vector Regression (SVR.rbf), and Bayesian Ridge Regression (BR), was evaluated using k -fold cross-validation (CV). The definition is as follows:

$$CV = \sqrt{\frac{1}{n} \sum_{i=1}^n (y_i - \hat{y}_i)^2}$$

where n is the number of observations in the training dataset, y_i is the true value, and \hat{y}_i is the predicted value. In searching for suitable model hyperparameters, we traverse the given hyperparameter space through a grid search, train each set of possible hyperparameter combinations and evaluate their performance on the validation set to select the set of hyperparameters and find the best model.

4. Regression fitting: Firstly, the feature matrix and target variables are extracted from the given dataset. Next, a random forest regression model was used for training and predictions

were made on the test set. To assess the model performance, we calculated the coefficient of determination (R^2) and the mean absolute error (MAE) to evaluate the model fit.

5. Prediction: The new composition space is constructed based on the given elemental ratios and valence information, which needs to satisfy the sum of the valences of the A -site and B -site elements to be +6, and ultimately to meet the requirements for the formation of the perovskite structure ABO_3 . After screening out the material combinations that meet the requirements, they are transformed into the corresponding feature values matrix. Subsequently, a trained Random Forest model is used to predict the new data, and all the cross-validated predictions (for each fold) are averaged to obtain the final predicted values to reduce the bias of the model, thus providing guidance for the material design.

6. Dataset sample size reasonableness validation: In order to verify the effectiveness of a relatively small amount of data for machine learning, the learning curve of the optimized random forest model showing the training scores for different sample sizes in the training set is provided in Supplementary Figure 1. It can be seen that when the sample size in the training set exceeds 80, the validation set score reaches a high and stable level, and the model is not affected by overfitting or underfitting. This is a good indication that the model can learn effective information and make accurate predictions even with a relatively small amount of data.

7. Data and code available: The feature values for the dataset can be found in the supplementary data. The related codes have been uploaded to Code Ocean.

Revision: The optimization and methods of machine learning has been added in the Supplementary Information.

Comment 5: The rationale for selecting BNT as the base system and specific elements for doping is not explained.

Reply 5: Thanks for the comments. The chosen BNT-based system is based on the following reasons: (1) BNT-based system is the most intensively studied lead-free dielectric energy-storage among all the perovskite-structured oxide ferroelectrics, owing to its relatively large polarization. (2) Currently, BNT-based system presents best energy storage properties compared other systems, such as BT ($BaTiO_3$), AN ($AgNbO_3$), NN ($NaNbO_3$), ST ($SrTiO_3$)-

based, etc. It means that there is enough data available to be used for screening statistics for machine learning.

For the selection of dopant elements, we control the number of dopant elements to ensure that there are enough samples for each dopant element when setting up the dataset. Because if the number of samples is insufficient, it will result in the regression model not being able to learn enough about these elements, leading to a decrease in the prediction accuracy. Therefore, we excluded the doping elements with a small number of samples, and in order to ensure that the model can predict more accurately, we only did further screening from the 13 doping elements included in the dataset, and as we described on page 7 of the manuscript, by comparing the magnitude of the eigenvalues of the three most important features of these candidate elements, we finally screened out seven different *A*-site and *B*-site doping elements to predict and screen the new components of high-entropy dielectrics.

Revision: We have added reasons for choosing BNT as the base system and doping specific elements in the manuscript.

Page 5: BNT-based solid-solution system is the most extensively and intensively studied perovskite-structure system for dielectric energy-storage owing to its relatively large polarization and excellent energy-storage properties.^{7,8} Due to the more abundant data points of reported energy storage properties, the BNT-based system was selected for developing ML model. The small-scale dataset used for the ML model comprised 121 BNT-based energy-storage bulk ceramics from literatures, which includes chemical composition, E_B , and W_{rec} (Supplementary Table 1). In the ML process, 16 A/B-site elements extracted from the dataset and 60 different descriptors were adopted (Fig. 1 a,b and Supplementary Table 2).

Comment 6: The study applied different ratio constraints for various doping elements ($0 \leq x_{La} \leq 0.15$, $0 \leq x_{Sr} \leq 0.2$, $0 \leq x_{Ca} \leq 0.2$, $0 \leq x_{Ta} \leq 0.08$, $0 \leq x_{Zr} \leq 0.2$, $0 \leq x_{Mg} \leq 0.1$, $0 \leq x_{Hf} \leq 0.1$), but the reasons for these specific constraints are unclear.

Reply 6: Thank you for your comments. We would like to point that for the dopant elements, such as La, Ta, Zr, Mg and Hf, the ratio constraints set are in line with the range covered in the Group 121 dataset. It allows to match as much as possible the modelling established in order to make valid predictions. For Ca and Sr, it has been shown in the literature that the high content will lead to a significant decrease in the polarization value. Therefore, we appropriately narrowed the range of their ratio constraints, which are of course within the range covered by

the dataset. In this way, the predictions are optimized as much as possible while maximally fitting the model range. We also provide explanations in the corresponding sections of the manuscript.

Revision: We have added discussions of elemental ratio constraints to the manuscript.

Page 7: Based on the identified elements, the following ratio constraints were applied to the different doping elements by substituting and combining these elements: $0 \leq x_{La} \leq 0.15$, $0 \leq x_{Sr} \leq 0.2$, $0 \leq x_{Ca} \leq 0.2$, $0 \leq x_{Ta} \leq 0.08$, $0 \leq x_{Zr} \leq 0.2$, $0 \leq x_{Mg} \leq 0.1$, $0 \leq x_{Hf} \leq 0.1$, with variations in steps of 0.01. These ratio constraints are largely in line with the ranges covered by the corresponding elements in the dataset, which enables a better match the established model as a way to achieve effective predictions.

Comment 7: Grain size significantly influences ferroelectric properties and energy storage, yet there is no microstructure analysis of samples B, C, and D to confirm similar grain sizes among all samples.

Reply 7: Thank you for your comments. In the revision, we have conducted microstructure analysis on samples B, C, and D using SEM (Supplementary Fig. 4). It can be seen that all samples are relatively dense, and sample A has the smallest grain size.

Supplementary Fig. 4 a-c Scanning electron microscopy (SEM) image displaying the surface morphology and grain size distribution of sample B, C, D and **d** Comparison of average grain size for all samples.

Revision: The SEM results have been added to Fig. 4 of the Supplementary Information, and

corresponding descriptions have also been included in the manuscript.

Page 8: “The microstructures of high-entropy sample A were analyzed (Fig. 2). Sample A was sintered well, with fine and homogeneous grains of about 0.8 μm (Fig. 2a), which representing the smallest grain size compared to samples B, C, and D (Supplementary Fig. 4).”

Comment 8: The authors optimized samples A, B, C, and D by comparing the largest maximum polarization value at 10 kV/mm and bandgap, but why not directly compare the energy storage capacity and breakdown electric fields of these samples?

Reply 8: Thanks for the comment. We would like to point that measuring the P - E loops at low electric fields and the bandgap of bulk ceramics is much easier and more convenient compared to directly measuring the energy storage capacity and breakdown electric fields. This is because it usually takes a significant amount of time to polish the ceramics samples down to about 50 μm and made special Au electrodes for energy storage performance test. Therefore, we can use the easy-measurements of P - E loops at low electric fields and the bandgap to further screen the compositions predicted by machine learning.

In the revision, we also tested the unipolar P - E loops of samples B, C, and D under high electric field and calculated their energy storage properties (Supplementary Fig. 9). The results demonstrate that sample A has the highest P_m at low electric field and still maintains the highest value at high electric field. Additionally, sample A, with the highest band gap, has the highest E_B and the best energy storage performance. These results are consistent well with the measured P - E loops at low electric fields and the bandgap. Overall, we can used more simple and convenient measurements to further screen the compositions predicted by machine learning.

Supplementary Fig. 9 Unipolar P - E loops measured under breakdown field of sample A, B, C and D.

Revision: The unipolar P - E loop have been added to Fig. 9 of the Supplementary Information.

Page 13: Notably, the energy-storage performance of samples B, C and D was also evaluated (Supplementary Fig. 9). As anticipated, the sample A exhibits the best energy-storage performance among the four compositions, and is consistent well with the results from the basic and facile experiments. It indicates that the compositions predicted by ML can be further refined effectively.

Comment 9: The claim that "smaller grain size generally implies higher breakdown field strength" needs supporting evidence.

Reply 9: Thank you for your comments. It has been well-demonstrated in the literatures that the breakdown strength of ceramics increases with decreasing grain size, and there is an exponential decay relationship with grain size (*J. Eur. Ceram. Soc.*, 21, 389-397, 2001; *J. Am. Ceram. Soc.*, 100, 3599-3607, 2017). This is because a smaller grain size means a higher density of grain boundaries, and the hole space charge layer accumulates at the grain boundaries, forming a barrier to carrier transport and leading to high resistivity (*Ferroelectrics*, 133, 109-114, 1992). Therefore, smaller grain sizes generally imply higher breakdown field strength.

Revision: The explanation that smaller grain sizes imply higher breakdown field strengths is given in the manuscript, and the relevant references are introduced.

Page 8: "1) The observed ultrahigh E_B can be attributed to several factors: 1) A refined grain size of 0.8 μm , as smaller grain size generally implies higher breakdown field strength.⁶⁰ The breakdown strength of ceramics increases with decreasing grain size, as small grain sizes increase the density of grain boundaries, and the hole charge layer accumulates at the grain boundaries, forming a barrier and leading to high resistivity;⁶¹"

Comment 10: The horizontal axis in Fig. 4c lacks a label.

Reply 10: We thank the reviewer for point this error. Now the label has been added.

Revision: Fig. 4c has been updated.

We thank reviewers again for the revision suggestions and all above useful questions and comments. Hopefully the manuscript has been improved by taking into account all above comments and addressing all above questions.

With best regards,

Sincerely yours,

Prof. Dr. Jun Chen on behalf of all authors

Beijing Advanced Innovation Center for Materials Genome Engineering

University of Science and Technology Beijing

Beijing 100083, China.

Tel./Fax: +86-10-62332525

Email: junchen@ustb.edu.cn

Dear reviewers,

Thank you so much for spending your time reading our manuscript and providing insightful comments. In the following, we present our response to your comments. We have highlighted the change within the manuscript in the blue-colored text.

Reviewer #1 (Remarks to the Author):

I am happy that authors has put great efforts for the improvement of the manuscript.

However, unit of dielectric breakdown strength reported in supplementary table 9 is kV/cm while the unit used in elsewhere is kV/mm. Here is the confusion. If I believe that unit kV/mm shown in the figure 3a is correct, then break down strength of the reported bulk ceramic to be around 950 kV/cm which is hard to imagine in the bulk ceramic. Author should clarify it and be consistent throughout the manuscript.

Reply: We are very grateful to the reviewers for recognizing our work and pointing out this error. In the Table 9 of the Supplementary Information, the unit of E_B should be kV mm⁻¹. We have clear this mistake.

Correspondingly, the unit of electric field used in the the manuscript correct. In this work, the breakdown strength of sample A is 950 kV cm⁻¹ (95 kV mm⁻¹), which is a relatively high value among the bulk energy-storage dielectric ceramics (Fig. 3d). It is mainly due to its small grain size and high resistivity, as well as its high band gap and Vickers hardness. Notably, for the energy-storage measurements, the ceramic is typically polished into thick ceramic plate with thickness around 50 μm. A dielectric breakdown strength of 1000 kV cm⁻¹ is equivalent to applying a voltage of 5 kV on a 50 μm-thick ceramic plate. As shown the Fig. 3d, in some energy-storage dielectric ceramics, the breakdown strength can reach 1000 kV cm⁻¹ or even higher (*Chem. Eng. J.* 478, 147383, 2023). For ferroelectric thin film materials with thinner thicknesses, the breakdown strength can reach to 5000 kV cm⁻¹ (*Nat. Mater.* 21, 1074-1080, 2023).

Revision: We have corrected the unit of electric field into kV mm⁻¹ in Table 9 of the revised Supplementary Information.

Supplementary Table 9. Comparison of the energy-storage properties in terms of W_{rec} , η , and E_B between sample A and reported lead-free bulk ceramics.

System	W_{rec} (J·cm ⁻³)	η (%)	E_B (kV·mm ⁻¹)	Ref.
	0.684	87.5	12.9	1

Reviewer #2 (Remarks to the Author):

After evaluating the changes made by the authors in response to the reviewers' comments, this referee finds that most aspects of the study have been addressed. The authors have tried to address the previous comments and questions that are crucial for the readers to make deeper understanding that they claim to demonstrate. The work is suited for publication in a journal with the scope of Nature Communications.

Reply: We sincerely appreciate the reviewer for reviewing our manuscript and agreeing to publish.

With best regards,

Sincerely yours,

Prof. Dr. Jun Chen on behalf of all authors

Beijing Advanced Innovation Center for Materials Genome Engineering

University of Science and Technology Beijing

Beijing 100083, China.

Tel./Fax: +86-10-62332525

Email: junchen@ustb.edu.cn

Reviewer #1 (Remarks to the Author):

Authors have corrected the error and revised the manuscript. Now the article can be considered for publication in Nature Communications.

Reply: We deeply appreciate you taking the time to review our manuscript and agreeing to publish it.

With best regards,

Sincerely yours,

Prof. Dr. Jun Chen on behalf of all authors

Beijing Advanced Innovation Center for Materials Genome Engineering

University of Science and Technology Beijing

Beijing 100083, China.

Tel./Fax: +86-10-62332525

Email: junchen@ustb.edu.cn